# Efficacy of maternal $B_{12}$ supplementation in vegetarian women for improving infant neurodevelopment: protocol for the MATCOBIND multicentre, double-blind, randomised controlled trial

Jitender Nagpal,[1] Manu Raj Mathur,[2] Swapnil Rawat,[1] Deepti Nagrath  ,[2] Charlotte Lee  ,[3] Atul Singhal,[3] Michelle Heys,[3,4] Mario Cortina Borja,[3] Katrin Augustin,[3] Jageshwor Gautam,[5] Rajendra Pant,[5] Laura Swabey  ,[3] Monica Lakhanpaul  [3,6]

For numbered affiliations see end of article.

**Correspondence to**
Professor Monica Lakhanpaul;
m.lakhanpaul@ucl.ac.uk

## ABSTRACT

**Introduction** Vitamin $B_{12}$ deficiency is widely prevalent across many low- and middle-income countries, especially where the diet is low in animal sources. While many observational studies show associations between $B_{12}$ deficiency in pregnancy and infant cognitive function (including memory, language and motor skills), evidence from clinical trials is sparse and inconclusive.

**Methods and analysis** This double-blind, multicentre, randomised controlled trial will enrol 720 vegetarian pregnant women in their first trimester from antenatal clinics at two hospitals (one in India and one in Nepal). Eligible mothers who give written consent will be randomised to receive either 250 mcg methylcobalamin or 50 mcg (quasi control), from enrolment to 6 months post-partum, given as an oral daily capsule. All mothers and their infants will continue to receive standard clinical care. The primary trial outcome is the offspring's neurodevelopment status at 9 months of age, assessed using the Development Assessment Scale of Indian Infants. Secondary outcomes include the infant's biochemical $B_{12}$ status at age 9 months and maternal biochemical $B_{12}$ status in the first and third trimesters. Maternal biochemical $B_{12}$ status will also be assessed in the first trimester. Modification of association by a priori identified factors will also be explored.

**Ethical considerations and dissemination** The study protocol has been approved by ethical committees at each study site (India and Nepal) and at University College London, UK. The study results will be disseminated to healthcare professionals and academics globally via conferences, presentations and publications. Researchers at each study site will share results with participants during their follow-up visits.

**Trial registration number**
CTRI/2018/07/015048 (Clinical Trial Registry of India); NCT04083560 (ClinicalTrials.gov)

### Strengths and limitations of this study

► Large-scale, multicentre, randomised controlled, double-blind trial across a low- and middle-income country and low-income country to determine the efficacy of maternal vitamin $B_{12}$ supplementation during pregnancy on neurodevelopmental outcomes 6 month post-partum.

► Study design benefits from the preceding qualitative patient and public involvement and nested qualitative substudy, ensuring participant feedback and process evaluation to optimise trial conduct.

► The study is being conducted across two diverse socio-demographic populations to increase generalisability of study findings.

► Normal/routine care differs at the two sites, which could lead to differences in the conduct and monitoring of the trial.

► Health system regulatory structures differ in the two countries.

## INTRODUCTION

Malnutrition during pregnancy is a global health problem. Micronutrient deficiencies are associated with adverse developmental outcomes in children, including poorer neurocognitive functioning which can potentially lead to poorer educational achievement and economic prospects.[1 2]

Vitamin $B_{12}$ deficiency is prevalent across many low- and middle-income countries (LMICs).[3–6] Partaking in a vegetarian diet, or one low in animal food sources, is a key risk factor in developing $B_{12}$ deficiency.[7 8] Sufficient $B_{12}$ intake during pregnancy is vital for fetal brain development.[9] Deficiency of $B_{12}$ in pregnancy is linked to infant birth defects, intrauterine growth restriction, preterm

**BMJ**

delivery and neural tube defects.[10–14] Many observational studies have shown associations between $B_{12}$ deficiency during pregnancy or early infancy with subsequent poorer infant neurodevelopment[15] and cognitive function[16 17] including memory,[18] language and motor skills.[19]

There is limited evidence from randomised controlled trials (RCTs) regarding the effect of vitamin $B_{12}$ supplementation and infant cognitive outcomes.[20–22] An infant supplementation trial observed neurodevelopmental benefits of infant $B_{12}$ supplementation with folate, and documented 0.45 (95% CI 0.19 to 0.73) and 0.28 (95% CI 0.02 to 0.54) higher SD-units in the domains of gross motor and problem-solving functioning, respectively.[22] However, in this study there was no significant benefit of giving vitamin $B_{12}$ or folic acid alone on early child development (with the exception of vitamin $B_{12}$ in relation to gross motor functioning[22]). In another trial, maternal vitamin $B_{12}$ supplementation given as 50 mcg a day from <14 weeks' gestation through to 6 weeks post-partum did not improve infant neurodevelopmental outcomes at 9 months.[21] In this trial hyperhomocysteinaemia (a marker of $B_{12}$ insufficiency) during pregnancy was negatively associated with infant cognitive outcomes. The study documented a deterioration in biochemical $B_{12}$ status in the placebo group, while the rise in $B_{12}$ status in the supplemented group (50 mcg) was inconsistent (no significant difference between first and third trimester levels). Additionally, a high rate of participant attrition (51.4%), poor monitoring of supplement compliance, a lack of biochemical improvement in $B_{12}$ status in the supplemented group of mothers and lack of infant biochemical evaluation for $B_{12}$ status in later infancy, further limit the inferences that can be drawn from this trial. Other studies that assessed a 250 mcg dose of $B_{12}$ per day throughout pregnancy until 3 months' post-partum in Bangladesh,[23] and >2000 mcg from 17 to 34 weeks' gestation in India[24] did not evaluate neurodevelopment outcomes in infancy. Hence, there is insufficient evidence concerning timing, dose, duration and efficacy of $B_{12}$ supplementation with reference to infant neurodevelopment.

To address this evidence gap, we are conducting a double-blind, multicentre, RCT in one LMIC (India) and one low-income country (Nepal), which will evaluate the efficacy of 250 mcg/day vitamin $B_{12}$ supplementation in vegetarian mothers from the first trimester to 6 months' post-partum on infant neurodevelopmental outcomes at age 9 months. Although, several published protocols report ongoing work on $B_{12}$ supplementation in India and Nepal,[25 26] the proposed work differs significantly in several ways.

First, we specifically chose to supplement vegetarian women, as they are a high-risk population due to their low $B_{12}$ intake, and as such have a high prevalence of $B_{12}$ deficiency. Second, an interventional dose of 250 mcg was chosen because it has been shown in an earlier trial to improve $B_{12}$ levels in pregnancy[23] (although infant neurodevelopment was not assessed in this trial). Third, we chose 50 mcg $B_{12}$ per day as a quasi-control group as this

dose has been shown to prevent deterioration in $B_{12}$ status during pregnancy in India.[21] Furthermore, given the progressive decline in $B_{12}$ levels throughout pregnancy (documented in the placebo group of earlier trials[21 22]), and the potentially serious neonatal consequences of severe $B_{12}$ deficiency highlighted above, a pure placebo control was thought to be unethical.

## AIMS AND OBJECTIVES
### Primary objective
To determine the effect of 250 mcg versus 50 mcg daily oral vitamin $B_{12}$ supplementation of pregnant women from the first trimester to 6 months post-partum on infant neurodevelopment at age 9 months.

### Secondary objectives
To determine the effect of 250 mcg versus 50 mcg of oral vitamin $B_{12}$ supplementation in pregnancy and 6 months post-partum on biochemical parameters of $B_{12}$ status (vitamin B12 levels, homocysteine and holotranscobalamin) in mothers at the end of the third trimester and in infants at age 9 months.

### Tertiary objectives
1. To evaluate the influence of the following effect modifiers:
   a. Socio-economic indicators (income, education, profession) on the relationship between maternal $B_{12}$ status, supplementation and infant neurodevelopment;
   b. Type of milk feeding on the infant vitamin $B_{12}$ status and infant neurodevelopment at 9 months;
   c. Maternal dietary $B_{12}$ intake with maternal and infant $B_{12}$ status, supplementation and infant neurodevelopment;
   d. Infant complementary feeding with infant vitamin $B_{12}$ status and infant neurodevelopment at 9 months;
   e. Maternal and infant $B_{12}$ status and neurodevelopment at 9 months after birth.
2. To determine the interaction of maternal iron and vitamin D status with the relationship between infant vitamin $B_{12}$ status and infant neurodevelopment at 9 months.

## METHODOLOGY
### Trial registration
The study is registered as a primary clinical trial at the WHO International Clinical Trials Registry Platform through the Clinical Trial Registry of India (CTRI/2018/07/015048) and at ClinicalTrials.gov (ID NCT04083560).

### Setting
This study includes two sites: Delhi, India and Kathmandu, Nepal. Both countries have documented a high prevalence of $B_{12}$ deficiency.[5 6]

### Participants
Mothers will be recruited from the (1) Sitaram Bhartia Institute of Science and Research (SBISR), New Delhi,

India, a privately-funded, tertiary hospital catering for a high- and middle-income population, with approximately 700 deliveries annually; and (2) Paropakar Maternity and Women's Hospital (PMWH), Kathmandu, Nepal, a government hospital catering for a poorer socio-economic area with approximately 22 000 deliveries annually. Our earlier work (2011 to 2014) documented that the majority of mothers delivering at SBISR were well-educated (96.8% college education), employed (63.2%) living in high income (annual income 16 000 US$) nuclear families (57.9%).[27] A directly comparable data set was not available from PMWH.

## Feasibility data and preliminary work

Feasibility of data and preliminary qualitative work was conducted for the trial in 2017 to 2018. Feasibility of recruitment within the given timelines was studied on the basis of previous data available from the study sites. The lead author (JN) examined neurodevelopmental outcomes of 413 babies born at SBISR in a cohort study conducted between 2011 and 2014. The project had a similar design to that proposed for this trial, using antenatal recruitment and post-delivery infant neurodevelopmental follow-up.[27] Approximately two-thirds (67%) of subjects were retained to follow-up in the cohort study. The loss to follow-up (33%) and speed of recruitment have been considered in the sample size calculations and the recruitment timeline for this trial. No similar recruitment and retention data were available from PMWH.

## Patient and public involvement

To represent the participants' perspective, a patient and public involvement (PPI) panel was established consisting of four parent pairs (two pairs per site) who had delivered a baby within the previous 2 years. One parent pair has continued to be involved with the project development and the Trial Oversight Committee. We conducted an exploratory study with pregnant mothers at both study sites to better understand the factors likely to influence the uptake and implementation of the proposed intervention, and proposed barriers to implementation (data unpublished). This qualitative study established that most respondents were unaware of $B_{12}$ dietary sources and the importance of $B_{12}$. All the participants involved in this exploratory study understood the burden of the intervention and were eager to know about symptoms, treatment and prognosis of $B_{12}$ deficiency. Respondents were also willing to accept the intervention and the need for drawing extra blood from the baby as a part of routine sampling for $B_{12}$ levels. Perceived barriers to the study included concerns around potential side effects of the $B_{12}$ supplement, relocation mid-trial and financial difficulties. All of these factors were considered in the study design, development of patient information, consultation and follow-up procedures as follows:

► All participants at the two study sites will be informed of the role, sources and importance of vitamin $B_{12}$ in human health using educational banners and posters and through community workshops;
► The initial information about the trial will be given by the doctor providing antenatal care to the mother. Subsequent detailed consent and recruitment will be conducted by study research staff;
► In order to improve compliance and study retention, reminder text messages will be sent weekly to participating mothers;
► To avoid the risk of financial loss to participants, various biochemical tests at the Indian trial site will be sponsored, and transportation costs at the Nepal site will be covered;
► Bradleys' Home Observation for Measurement of the Environment (HOME) inventory was added to the assessment given the divergent environmental profile of the two study sites,[28] in order to assess the infant's environment and possible environmental confounders on neurodevelopment.

The study results will be disseminated globally to academics and health professionals via conferences, presentations and publications. Researchers at each study site will share results with participants during their follow-up visits.

Also, in preparation for this trial we adapted a prevalidated Food Frequency Questionnaire (FFQ) for vitamin $B_{12}$ intake[29] to the Indian and Nepalese settings. In brief, the original vitamin $B_{12}$ FFQ by Mearns[29] (including 30 questions on consumption of food and beverages containing vitamin $B_{12}$) was reviewed independently by two nutritionists. A market survey was done to obtain information on egg and milk products consumption, as well as the availability of $B_{12}$ fortified items at local markets in Delhi and Kathmandu. Homemade recipes which included egg and/or dairy products as a main ingredient were incorporated into the FFQ. Measurements used in the FFQ were standardised based on the typical/locally available household utensils to ensure accurate responses and to maintain consistency across the sites.

All research staff from India and Nepal will be jointly trained in study design, research methodology, good clinical practice, the study protocol and study tools. Training lectures will be video-recorded and made available to the teams at both study sites as a reference material for continuous training.

## Eligibility

Potentially eligible participants will be screened at first presentation to the antenatal clinic at ≤12 weeks of gestation in accordance with the following criteria:

### Inclusion criteria

► Consuming a vegetarian diet – including veganism and/or people who do eat egg and/or do consume milk and/or a portion meat/chicken/fish <once a month;
► Living within an a priori defined geographical area;
  – Delhi – national capital region;

– Nepal – 10 km radius of PMWH;
► Able to understand at least one of the following languages – English, Hindi or Nepalese.

### Exclusion criteria
► Wish for medical termination of pregnancy;
► Aged <18 or >35 years;
► Already on medicinal vitamin $B_{12}$ supplementation (including B-complex or multivitamins);
► Multiple gestation;
► Diagnosis of a chronic medical condition (including diabetes mellitus, hypertension, heart disease, neurological disease or thyroid disease);
► Testing positive for hepatitis B, HIV or syphilis;
► Anticipating moving out of the a priori defined geographical area before or after delivery;
► Undergoing treatment for infertility;
► Having a known pre-diagnosed mental health disorder, including depression, drug or alcohol abuse;
► Currently participating or having participated in another study within 4 weeks prior to the trial commencing;
► Having a known allergy to vitamin $B_{12}$ or another supplement constituent.

### Consent
Eligible participants meeting the selection criteria will be given time to think over continuation of pregnancy, site of delivery, participation in the trial and any other considerations. They will be contacted within 48 hours by the research team regarding their decision to participate, and will be invited to provide written or video-recorded consent. Participants will be able to opt into the trial any time before 12 weeks of gestation.

### Randomisation
As presented in figure 1, all eligible and consenting participants will be randomised with an allocation ratio of 1:1 to either the intervention (250 mcg) or control (50 mcg) group. The randomisation sequence will be generated at a single, third-party location (non-study site) using a computer-generated sequence stratified for location. The sequence will be allocated to serial numbers by a neutral person in each country using sequentially numbered, sealed, opaque envelopes. Codes within envelopes will identify boxes labelled with five-digit codes. Each box will contain enough dosages to cater to the study period of one participant. The strips containing intervention and control doses of $B_{12}$ will be identical in taste, smell and appearance. All participants, recruiters, developmental therapists, laboratory staff and the data analyst will be blinded to the $B_{12}$ allocation until results of final analysis are made available.

### Procedure

### Upon enrolment
Upon enrolment, demographic and clinical details, including age, height, weight, ethnicity, education and socioeconomic status, will be collected. The participants' preferred channel of subsequent communication (phone, text message, email) will be noted. Participants will be allocated a study identification number and accompanied by the researcher for blood sampling. These samples will be taken at the same time as other samples being drawn in the first trimester to minimise unnecessary venepunctures. Twelve mL of blood sample will be drawn into two plain vials (5 mL each) and 2 mL will be drawn into an EDTA vial. The EDTA sample will be processed immediately for complete blood counts by coulter counter. The plain (serum separator tubes) vial samples will be centrifuged at 3500 rpm (Rotor radius 15 cm) for 10 min at the study site. The separated serum will be transported on ice and consistently stored at −70°C, and will be monitored at both the study sites. Participants will be given enough $B_{12}$ supplement doses to last 5 days beyond the expected date of their next visit, after being fully informed about the supplement and its administration. Participants will be contacted 48 hours after receiving the $B_{12}$ doses to confirm initialisation of supplementation and re-emphasise compliance.

### Subsequent maternal visits
Subsequent maternal visits will take place monthly until 36 weeks, and then weekly thereafter. At these visits, all participants will visit their obstetrician as part of routine care. Participants will also meet with the research staff at a prespecified location at the study site. To maximise compliance, weekly adherence and pre-appointment reminders will be sent via the prespecified preferred mode of communication. Acquired morbidity (gestational diabetes, hypertension and hypothyroidism) and medication history will be noted in case record forms. Empty supplement packets will be collected at each appointment as an indicator of adherence. In the third trimester, mothers' dietary $B_{12}$ intake will be assessed using a pre-standardised and adapted FFQ, and blood sampling will be repeated using the same procedures as in the first trimester.

### Neonatal assessment
During childbirth, a medical officer and/or paediatrician will monitor the birth and post-delivery health of neonates for any morbidity that might potentially influence neurodevelopment. Gestational age, growth retardation, congenital abnormalities and APGAR (appearance, pulse, grimace, activity, respiration) score will be routinely documented for neonates. Cord pH will be measured for all newborns with a delayed cry and who require any intervention beyond the basic steps of resuscitation (ambu bagging, intubation or pharmacological treatment). Any seizures, neurological problems, hypoglycaemia, hypothermia or hyperbilirubinaemia will be assessed, diagnosed, evaluated and managed as per institutional guidelines. As per standard practice, all newborns will be screened for hearing deficits, vision and critical congenital heart disease according to institutional protocols.

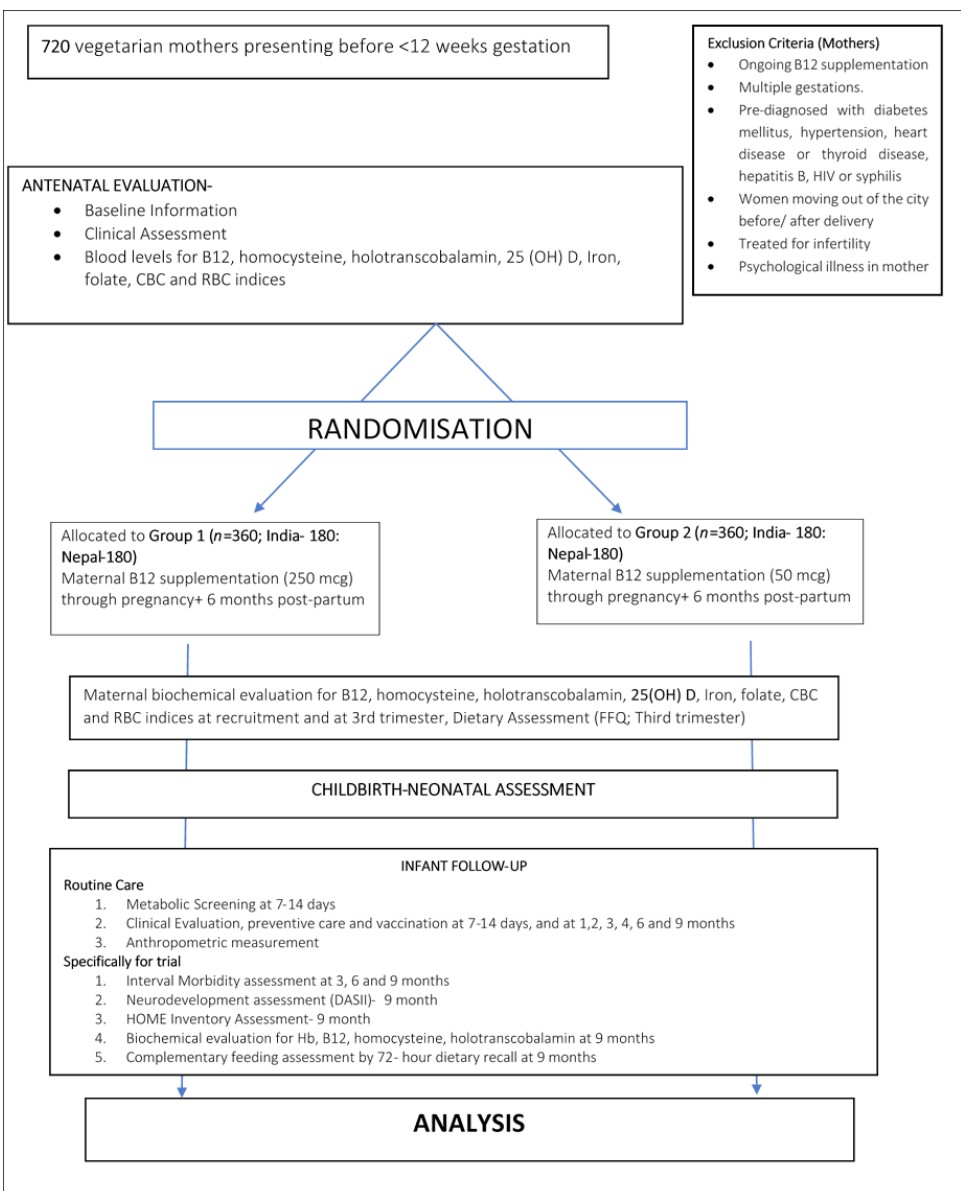

**Figure 1** Study design and participant flow through the study. CBC,complete blood count; DASII, Developmental Assessment Scales for Indian Infants; FFQ, Food Frequency Questionnaire; Hb,haemoglobin; HOME, Bradleys' Home Observation for Measurement of the Environment; RBC, red blood cell; 25(OH)D, 25-hydroxyvitamin D.

Preterm neonates transferred to the neonatal unit will be monitored as per their customised follow-up and early intervention plans, implemented as recommended by the paediatrics and/or primary care provider. Neurodevelopment and biochemical assessment of preterm neonates will be carried out as per their corrected gestational ages.

### Infant follow-up
Infant follow-up will be conducted at 7 to 14 days, 4, 8, 12 and 16 weeks and at 6 and 9 months of age. At these visits, the infants will undergo routine care as well as study specific evaluations (see Visit Schedule: online supplementary table 1).

### Routine care
Each mother-infant dyad will see a paediatrician for age-appropriate routine care, which will involve clinical evaluation, preventative care and vaccination as per standard procedures in each site. Anthropometric measurements and signs of micronutrient deficiency (anaemia and rickets) will be recorded during each visit as part of routine care. Metabolic disorders (congenital hypothyroidism, congenital adrenal hyperplasia, cystic fibrosis, biotinidase deficiency, phenylketonuria and galactosaemia) will be screened for and maternal support for breastfeeding by a paediatrician or lactation counsellor will be provided at 7 to 14 days.

### Study specific
Supplements will be provided to mothers at each visit, with instructions to take until 6 months after childbirth. Any morbidities and deficiencies will be recorded and treated as per institutional guidelines. Maternal and infant tolerance for the supplementation, including any gastrointestinal symptoms, will be recorded at each visit,

as will maternal compliance. Provision of supplements to the mother in both groups will be stopped at 6 months post-partum.

The hospital evaluation at 9 months (±14 days) will include neurodevelopmental assessment by a developmental therapist using the Developmental Assessment Scales for Indian Infants (DASII)—an adapted and validated version of the Bayley's Scale for Infant Development Third Edition[30]—and blood sampling for estimation of haemoglobin, vitamin $B_{12}$, homocysteine and holotranscobalamin concentrations. The DASII motor scale assesses gross and fine motor skills, while cognitive, personal and social skills development are assessed through the DASII mental scale. All neurodevelopmental assessments will be conducted by a trained developmental therapist at the institution in a soundproof room. In premature babies, corrected gestational age will be used to calculate developmental quotients. Additional assessments will include a complementary feeding assessment by a nutritionist using a 72 hours dietary recall, and a home environment assessment using the adapted Bradley HOME inventory[28] by a trained field worker.

### Compliance

Research staff will dispense blister packaging of 10 $B_{12}$ capsules and study specific bags for participants to store, carry and keep supplements in one place. In case of loss of supplement or inability to come for follow-up, participants will be able to contact the researcher at any time to request more supplements. The supplement packaging will be numbered with a serial number. To maximise compliance, weekly reminders will be sent through the preferred mode of communication as specified by the trial participants. Each participant will be provided with a bag to save empty packaging strips and instructed to carry the bag with her at the next visit to monitor compliance.

### Biochemical analysis

All saved samples will be processed as one batch after the completion of the study. The maternal samples will be used to determine the blood levels for vitamin $B_{12}$, homocysteine, holotranscobalamin, 25-hydroxyvitaminD (25(OH) D), folate, iron studies (serum iron, ferritin and total iron binding capacity), complete blood count (CBC) and red blood cell (RBC) indices in the first and third trimesters. Infant blood will be used for haemoglobin, vitamin $B_{12}$, homocysteine and holotranscobalamin measurements. Vitamin $B_{12}$ and 25(OH)D will be measured by enhanced chemiluminescenceimmunoassay (CLIA) (Johnson & Johnson, USA). Iron levels will be estimated by pyridylazo dye (Johnson & Johnson, USA). Folate will be analysed by enhanced CLIA (Beckman Coulter, USA) and CBC (Beckman Coulter, USA). RBC indices will be calculated by impedance (Beckman Coulter, USA). Homocysteine and holotranscobalamin will be analysed by chemiluminescent microparticle immunoassay.

### Outcomes

The following study outcomes will be investigated:

#### Primary outcome

▶ DASII scores for infants at 9 months (±2 weeks) post-partum.

#### Secondary outcome(s)

▶ Change in $B_{12}$ status (vitamin $B_{12}$ level, homocysteine and holotranscobalamin) of mother between first (≤12 weeks' gestation) and third trimester (≥27 weeks' gestation);
▶ $B_{12}$ status (vitamin $B_{12}$ level, homocysteine and holotranscobalamin) of infants at 9 months (±2 weeks) post-partum.

#### Exploratory outcome(s)

▶ Haemoglobin of infants at 9 months (±14 days) post-partum;
▶ Infant anthropometry including weight, height/ length and head circumference at 9 months after birth in all subjects.

### Sample size considerations

A planned sample of 720 pregnant women, 360 per study site, will be randomly assigned to experimental (n=360; n=180 in per study site) or control (n=360; n=180 in per study site) groups. This is based on the number of infants needed to complete the neurodevelopmental screen in each of the two randomised groups to detect a difference in development quotient (DQ) at 9 months (primary outcome) of two points (presumed mean DQ in this population of infants at 9 months of 96.9, SD 7.07 on basis of earlier work).[27] The difference of two DQ points was chosen because it is the smallest effect size thought to be clinically significant, and hence a null trial result is likely to exclude a clinically relevant effect of maternal B12 supplementation on infant cognitive function. The sample size has been inflated by 35% to account for attrition, and assumes 90% statistical power and a 5% two-sided significance level.

### Statistical analysis

The primary and secondary outcomes will be analysed using an intention-to-treat analysis by a blinded statistician. Comparability of participants assigned to study groups at enrolment will be assessed using tests of association for categorical variables and by fitting generalised linear models for continuous variables. These models will include interaction terms to explore conditional differences between the trial's arms. If necessary, transformations will be applied to the response variables to fulfil the model's assumptions. Covariates will include trial site, infant feeding, gender, feeding by gender, weight z-score at birth, weight gain between birth and 3 months and maternal body mass index. Z-scores for height and weight will be calculated according to the WHO[31] growth standards using the R language for statistical computing.[32]

Subgroup analysis will be conducted to examine the differential impact of $B_{12}$ supplementation in high versus low risk mothers and infants. Mothers identified as having pregnancy-related or other morbidity will be classified as high-risk (eg, including gestational diabetes, pregnancy-induced hypertension/pre-eclampsia/eclampsia and absent or reversed end diastolic flow); and low-risk. Babies will be defined as high risk if there is any combination of the following:

► Small for gestational age, defined as <10th centile for birth weight;
► Very preterm (28 to 32 weeks gestation); extremely preterm babies (<28 weeks) will be excluded from the analysis;
► Experience of a higher risk neonatal course defined as experience of APGAR score <7 at 1 min, hypoglycaemia, hyperbilirubinaemia or prolonged neonatal intensive care stay >7 days);
► Infants with documented neurological disease/anomaly/illness will be subgrouped separately (high risk) and compared with those with no disease/anomaly illness (low risk).

### Data management and dissemination

Data will be collected under the regulation of the data protection and management guidelines provided by the Governments of India and Nepal, and as per the UK Data Protection Act 2018. Data will be securely stored in an E-database with password protection. The E-database will be designed using Access. Live data entry will be conducted at the study sites and tested for accuracy using duplicate data entry at an offsite location at periodic intervals.

The study results will be disseminated globally with academic and health professionals via conferences, presentations and publications. Researchers at each study site will share emerging results with participants during their follow-up visits. Regular site meetings with press offices, short films, radio spots, periodic blog posts and Twitter and Facebook messages are also planned to strengthen the trial's online presence and reach young and middle-aged Internet users for whom the information may be of interest.

### Qualitative evaluation

A qualitative analysis of the information learnt during preliminary PPI work aimed to explore knowledge, attitudes and perceptions of pregnant women towards $B_{12}$ supplementation and child development and the potential barriers to implementation. Better understanding these domains across the two study sites informed the perceived need for the intervention, helped define patient-centric outcomes and allowed the evaluation of any differences between the two arms.

We used convenience sampling to recruit and interview 25 participants across both trial sites (SBISR 12; PMWH 13). We developed and piloted a semi-structured topic guide in collaboration with our PPI representatives serving on the Trial Steering Committee. The topic guide was designed in English and translated into either Hindi or Nepali. The topic guide was then back-translated into English to ensure consistency and accuracy of content. It was created with the intention to help direct the interview while remaining sensitive to unsolicited themes. We ensured that an interviewer with prior training and experience in qualitative methods facilitated the semi-structured interviews.

Data was collected in quiet rooms in each hospital, away from other patients and staff. Interviews were facilitated in English, Hindi or Nepalese, depending on the preference of the participant. All interviews were audio-recorded, transcribed, translated into English (where necessary) and then cross-checked against the original recording. The translated transcripts were then coded using the ATLAS.ti 7.2 software for qualitative analysis.[33] A research team member coded and categorised the data, and prepared an initial narrative. A report detailing the findings is in the process of being finalised, with input from the interviewers and wider research team.

### Ethical considerations

The study protocol (10th December 2018. Version: 8.0), informed consent form, participant information sheet, tool and other supportive materials have been approved by the Institutional Ethics Committee (IEC) at SBISR (FL/SBISR/IEC/2018–01); Public Health Foundation of India (TRC-IEC-380/18); PMWH (59-11-433); Nepal Health Research Council Ethical Review Board (Reg No. 761/2018) and the University College London Research Ethics Committee (10901/001). The investigators at each study site have submitted to, and where necessary, obtained approval from the above parties for all substantial amendments to the original approved documents.

The trial will be conducted in accordance with the principles and rules of the Declaration of Helsinki. Informed and written consent will be obtained from all participants by the research teams at each study site. The trial will comply with the UK Data Protection Act 2018, ensuring the participants' anonymity. The participants will be identified only by a participant identification number on all trial documents and electronic databases. Every effort will be made to arrange interviews and examinations at a time convenient to the participants, without causing much disruption in their daily routine. Study data will be stored securely and will only be accessible to trial staff and authorised personnel. The Principal Investigator at each study site will submit an annual progress report, end of trial notification and a final report to the Institutional Ethics Committee, host organisations and the funders.

We used the Standard Protocol Items: Recommendations for Interventional Trials checklist for writing our protocol.[34]

## DISCUSSION

A positive result from this study will have considerable impact on current and future policy initiatives, by underpinning current micronutrient supplementation policy in India and Nepal. The project will provide key insights to strengthen existing Indian governmental nutrition programmes such as 'National Iron Plus Initiative', 'Janani Suraksha Yojana' and 'Janani Shishu Suraksha Karyakaram'. Because of the high worldwide prevalence of maternal $B_{12}$ deficiency (74%), the study findings will also be highly relevant to maternal and neonatal health in nutritionally vulnerable and at-risk populations globally.

Using a 50 mcg quasi-control group limits the inferences that can be drawn from a negative result (neurodevelopment comparable between the two groups). The study outcome could produce several results, which could be interpreted as follows:

1. *Findings:* Improvement in $B_{12}$ parameters from first trimester to third trimester is statistically significant in both groups, and the change is statistically comparable between two groups for mother and baby. Substantial proportion of mother/infants in both groups improve from $B_{12}$ deficient to replete state.
   *Interpretation:* 250 mcg dose is as effective as 50 mcg in improving biochemical parameters and both doses result in comparable neurodevelopment. 250 mcg dosing offers no additional advantage over 50 mcg.

2. *Findings:* Improvement in $B_{12}$ parameters from first trimester to third trimester is significant in either 250 mcg alone or both groups, but 250 mcg causes higher rise (statistically significant) in $B_{12}$ biochemical parameters than 50 mcg for mother and baby.
   *Interpretation:* 250 mcg is more efficacious than 50 mcg in improving biochemical $B_{12}$ status. Since neurodevelopment is equivalent between two groups a higher dose of supplementation does not offer any neurodevelopmental advantage despite better biochemical $B_{12}$ status.

3. *Findings:* A lack of biochemical improvement in both groups, although unexpected, would be interpreted to mean poor efficacy of 50 mcg or 250 mcg oral $B_{12}$ in improving $B_{12}$ status of mother-infant pairs. Both groups continue to have a high proportion of mothers deficient in $B_{12}$ before and after supplementation.
   *Interpretation:* Factors affecting $B_{12}$ status in the current trial (including compliance and socio-demographics) would have to be carefully analysed. Further work would be required to study the determinants of oral $B_{12}$ efficacy in detail.

Many studies have evaluated the impact of varying vitamin $B_{12}$ intake (physiological dosage (2 mcg/day) to pharmacological dosage (up to 2000 mcg/day); supplemented/observed) in diverse population sets.[20–24 35] Our study specifically seeks to address the question of the optimum dosage for vegetarian pregnant women. As one of the few large-scale trials of $B_{12}$ supplementation in pregnancy, the results of this study would be generalisable to vegetarian expecting mothers. We aim to contribute to the evidence base on improving neurodevelopmental outcomes in vulnerable infants. While a positive outcome for the trial would help to develop an intervention programme to reduce developmental delay using vitamin $B_{12}$ as a supplementation in pregnant women, a negative or null result would also add to existing knowledge and lay the groundwork for future work on the subject—especially around utility, dose, duration and time of initiation of vitamin $B_{12}$ supplementation.

**Author affiliations**
[1]Pediatrics, Sitaram Bhartia Institute of Science and Research, New Delhi, Delhi, India
[2]Public Health Foundation of India, New Delhi, India
[3]UCL GOS Institute of Child Health, University College London, London, UK
[4]Specialist Children's and Young People's Services, East London NHS Foundation Trust, London, UK
[5]Obstetrics and Gynaecology, Paropakar Maternity & Women's Hospital, Kathmandu, Nepal
[6]Community Paediatrics, Whittington NHS Trust, London, United Kingdom

**Acknowledgements** CL and ML were partly supported by the National Institute for Health Research Collaborations for Leadership in Applied Health Research and Care North Thames at the time of this work. We would also like to thank our Patient and Public Involvement panel. MH is funded by East London National Health Service Foundation Trust.

**Contributors** JN, MRM and ML conceived the study and AS, MH, MCB, JG, RP, ML and CL contributed to the study design. SR and DN conducted the qualitative study, adaptation of Food Frequency Questionnaire, standardisation of questionnaires and organised the training workshop. SR, DN, KA, LS and CL drafted the manuscript. JN, MRM and ML finalised the manuscript, and all authors assisted in developing the protocol and have read, reviewed, edited and approved the final manuscript.

**Funding** The study is funded by the Newton Fund: The Department of Biotechnology (DBT) India, in collaboration with Department of International Development (DFID), the Economic and Social Research Council (ESRC) and the Medical Research Council (MRC) consortium (Grant Number MR/R020396/1). This research was supported by the National Institute for Health Research (NIHR) Biomedical Research Centre based at UCL Great Ormond Street Institute of Child Health/Great Ormond Street Hospital NHS Foundation Trust.

**Competing interests** None declared.

**Patient and public involvement** Patients and/or the public were involved in the design, or conduct, or reporting or dissemination plans of this research. Refer to the Methods section for further details.

**Patient consent for publication** Not required.

**Provenance and peer review** Not commissioned; externally peer reviewed.

**ORCID iDs**
Deepti Nagrath http://orcid.org/0000-0003-3854-1841
Charlotte Lee http://orcid.org/0000-0001-8252-9538
Laura Swabey http://orcid.org/0000-0001-6677-1420
Monica Lakhanpaul http://orcid.org/0000-0001-5288-3325

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
