## [Reviewer comments · BMJ Open]

ARTICLE DETAILS

TITLE (PROVISIONAL)	Efficacy of maternal B-12 supplementation in vegetarian women for improving infant neurodevelopment: Protocol for the MATCOBIND multicentre, double-blind, randomised controlled trial
AUTHORS	Nagrath, Deepti; Nagpal, Jitender; Mathur, Manu; Rawat, Swapnil; Lee, Charlotte; Singhal, Atul; Heys, Michelle; Cortina Borja, Mario; Augustin, Katrin; Gautam, Jageshwar; Pant, Rajendra; Lakhanpaul, Monica

VERSION 1 – REVIEW

REVIEWER	Tor Strand Innlandet Hospital Trust, Norway
REVIEW RETURNED	03-Nov-2019

GENERAL COMMENTS	Well written protocol paper with excellent descriptions of the interventions and outcomes of an RCT of very high vs. high doses of methyl-cobalamin to pregnant vegetarian women in Kathmandu (Nepal) and New Delhi (India). I think, however, that the manuscript would have benefitted from another round of proofreading before submitting it to BMJ-OPEN. The introduction provides an excellent rationale for the study and for using Early Child Development as the primary outcome. The literature review includes studies in pregnant and infants and justifies the research question well. I have, however, the impression that the selected literature has not been updated. There is recent evidence from both observational and intervention studies on the relationship between vitamin B12 and neurodevelopment both from both Nepal and India. These studies can also shed some light on the value of plasma total homocysteine as a biomarker for B12 in pregnancy. I would, therefore, perform a quick search to update the information in the introduction. I would also check the reference list for errors. For example, reference no 18 has misspelled the name of the first author. Furthermore, the study described in reference 11 has published newer and more recent data relevant to this project. There is also another protocol paper published in BMJ-Open that describes a similarly designed RCT, however, with a somewhat larger sample size but with lower doses of cobalamin and that is currently being undertaken in Nepal. I agree that the study described in reference 11 has many methodological shortcomings and a rather high “attrition rate.” However, I disagree with the statement that there was a “lack of biochemical evaluation of B-12 status in the supplemented group”. One of the strengths of this study was the comprehensive
---

	biochemical evaluation of those who actually completed the study. Please modify the introduction. A central premise for the study is that 50µg is “almost” equivalent to a “pure” placebo as the primary outcome is to compare neurodevelopment between this high dose group and the very high dose group. Furthermore, the authors also claim that their findings will have a “considerable impact on current and future policy initiatives focusing on micronutrient supplementation in India and Nepal.” There is, however, a possibility that the control group (50µg methyl-cobalamin) also will benefit from daily supplementation of this dose of B12, which is, in fact, approximately 20x of the RDA. This design does not disqualify the purpose of the study but challenges the interpretation of potential null-findings. (Was it because they did not need the additional cobalamin, or because giving 50µg daily is as effective as giving 250µg). Related to this, the authors listed many shortcomings (than dose) of other similar studies in their literature review, which could explain the lack of effect. Please update the discussion regarding how null findings will be interpreted. The direct and indirect biomarkers of B12 will be crucial in the interpretation of their results, particularly negative findings. The description of the laboratory procedures is weak. Please improve this section. They should mention the planned biomarkers for estimating iron status. At this stage, I believe that the study has already started enrolment, the authors should accordingly know more about how and where the samples are going to be analysed and provide more details about the preanalytical sample processing and storage. Please update the manuscript with this information. The table, “Table 2. Blood samples and measurements” are incomplete and offer very little information about the biomarkers, biochemical methods, or sample. I am also unable to find a reference to this table in the text. It is positive that they mention the form of cobalamin that will be given (methyl-cobalamin) Inclusion criteria, exclusion criteria, and other central definitions are also clearly defined and makes sense. Please be clear in the objectives, whether it is the DAS1 or BSID that will be the variable used to address the main objective. Several additional objectives are listed. Please refrain from using the term “effect” when referring to analyses that will be undertaken in an observational design. I suggest that you include a section that describes the chosen doses of B12 in relation to the recommended daily intakes. Please ensure that the information provided in this manuscript (especially inclusion criteria, exclusion criteria, objectives, and outcomes match the information that is given in the clinical trial registries.
--	---

	Reconsider all the statements regarding strengths and limitations. This includes: “First large-scale, multi-centre randomised controlled, double-blind trial to determine efficacy of vitamin B-12 supplementation up to 6 months post-partum in improving neurodevelopmental outcomes in infancy.” There is a larger study with a similar objective going on in Nepal now The primary outcome is going to be measured at 9 months, why “up to 6 months post-partum” in this statement Minor issues Blood/plasma/serum “B12 levels” is very unscientific and imprecise. I would instead use the term “concentration” and “cobalamin.” I would also use the term total homocysteine rather than “homocysteine.” Metabolic screening, what is included in this assessment It is unclear to me whether or not the Bayley Scales of Infant development will be used or not. It is mentioned in the text and figure 1. Reconsider using reference 1 for gestational malnutrition
--	---

REVIEWER	Ingrid Kvestad Regional Centre for Child Mental Health, NORCE Norwegian Research Centre, Bergen, Norway
REVIEW RETURNED	06-Nov-2019

GENERAL COMMENTS	The protocol paper describes a multi-centre randomized controlled, double-blind trial to determine the efficacy of vitamin B12 supplementation up to 6 months post-partum in improving neurodevelopmental outcomes in infancy. Vegetarian mothers will be randomized to receive either 250 mcg or 50 mcg vitamin B12 daily, and the main outcome is neurodevelopment measured by an Indian adapted assessment tool the DASII. The paper is well written, extensive and in coherence with the information registered in clinical trials. There are some information lacking however on the rationale of the study;  - The study will only include vegetarian pregnant mothers. What is the rationale? What are the strengths/limitations of such a design? In my understanding, that the study is restricted to vegetarian pregnant women is such an important point of the study design, that it should be stated throughout the manuscript, including the title. - What is the rationale for the vitamin B12 doses? Why compare to 50mcg and not a full placebo? What are strengths/limitations to such a design. Both points should be addressed more closely in the paper, starting from (title) the introduction, objectives, methods and discussion. Some more comments to the manuscript: Introduction:
--

	The references used in the introductions seems outdated. There are more recent published reviews and studies on B12, perinatal health, children and cognition that would improve the introduction. Among these studies from New Delhi and the Kathmandu valley that should be very relevant for this study. Check also the literature on B12 and growth, both observational results from Nepal that are interesting for this study, and results from interventional trial in young children in Delhi. First sentence of third paragraph does not make completely sense, how can evidence be inconclusive when only one trial is referred to? Looks like there is some studies suggesting that there is a link, but very few (one?) trials have tried to confirm the effect of B12 on offspring neurodevelopment. In this trial however, the results do not provide evidence for a link between B12 and early child development. This could be due to dose? I would work some more on this paragraph. Page 4, lines 50 on the evidence gap: Should update to include that you will include only vegetarian pregnant women, and that you will compare doses of 250 to 50. In addition, I would use some effort in the introduction to build up to these essential elements of your study. Aims and objective: Primary objective: include vegetarian. Secondary objectives: please state which biochemical parameters. Tertiary objectives; please reconsider the wording "effect" – will the assessment of causality be possible in these objectives, or will you look for associations? Methodology Again, it is important to include the point that you are including vegetarians only, under subheading "participants": you state the number of births in Delhi and Kathmandu – how many of these mothers will be vegetarians? In my experience, not all. Procedure: check sentence line 20 page 12 Neurodevelopment assessment: quality control procedures such as training, standardization and prevention of examiners drift should be detailed. How to make sure that testing in Nepal and India is done in a similar manner for the scores to be comparable? Will cultural adaptations be necessary when using the DASI in Nepal? The HOME: when, how and by whom? This information is lacking. Statistics: Not clear to me how the subgroup analyses will be done? What are the subgroups. 2 subgroups, or 5 subgroups (bullet points), the sentence is not clear. Discussion Some more on strength and limitations both of including vegetarians, and on not using a placebo control. Generalizability. What would be the implication for the findings? Implications of null findings?
--	--

	Figure 1. Very clear. BSID III should be explained in a footnote. Please check the version, I think DASI is a version of the Bayley 2nd version, not the third. Also, I think you should also state DASI in the figure, not only BSID.
--	--

VERSION 1 – AUTHOR RESPONSE

Reviewers Comments	Author Response
Reviewer 1	
The introduction provides an excellent rationale for the study and for using Early Child Development as the primary outcome. The literature review includes studies in pregnant and infants and justifies the research question well. I have, however, the impression that the selected literature has not been updated. There is recent evidence from both observational and intervention studies on the relationship between vitamin B12 and neurodevelopment both from both Nepal and India. These studies can also shed some light on the value of plasma total homocysteine as a biomarker for B12 in pregnancy. I would, therefore, perform a quick search to update the information in the introduction. I would also check the reference list for errors. For example, reference no 18 has misspelled the name of the first author. Furthermore, the study described in reference 11 has published newer and more recent data relevant to this project. There is also another protocol paper published in BMJ-Open that describes a similarly designed RCT, however, with a somewhat larger sample size but with lower doses of cobalamin and that is currently being undertaken in Nepal.	Available literature on the subject has been thoroughly re-reviewed and some of the ongoing work has been added in the revised manuscript. Referencing has been rechecked and rectified. (Page 4-5)
I agree that the study described in reference 11 has many methodological shortcomings and a rather high “attrition rate.” However, I disagree with the	The reviewer has correctly highlighted that biochemical evaluation was conducted in the Srinivasan trial. However, there was no

statement that there was a “lack of biochemical evaluation of B-12 status in the supplemented group”. One of the strengths of this study was the comprehensive biochemical evaluation of those who actually completed the study. Please modify the introduction.	improvement in maternal vitamin B-12 status from the first to third trimester. In the supplemented arm increase was 3.0 (-54.8, 83.5) pmol/L which was not significant. No significant group differences in maternal MMA, tHcy, or prevalence of anaemia (hemoglobin <11g/dL) were noted at the second or third trimester time points. Also, the B-12 status of a subset of infants was only measured at 6 weeks while the neurodevelopment is assessed at 9 months. The language in the manuscript has been modified to better reflect this limitation. (Page 4-5)
A central premise for the study is that 50µg is “almost” equivalent to a “pure” placebo as the primary outcome is to compare neurodevelopment between this high dose group and the very high dose group. Furthermore, the authors also claim that their findings will have a “considerable impact on current and future policy initiatives focusing on micronutrient supplementation in India and Nepal.” There is, however, a possibility that the control group (50µg methyl-cobalamin) also will benefit from daily supplementation of this dose of B12, which is, in fact, approximately 20x of the RDA. This design does not disqualify the purpose of the study but challenges the interpretation of potential null-findings. (Was it because they did not need the additional cobalamin, or because giving 50µg daily is as effective as giving 250µg). Related to this, the authors listed many shortcomings (than dose) of other similar studies in their literature review, which could explain the lack of effect. Please update the discussion regarding how null findings will be interpreted.	The authors submit that for this study it is hypothesised that the lower dose (50 mcg) in actually a quasi-control group. This dose is specifically chosen as it proved ineffective in improving B-12 status while preventing deterioration in deficiency in pregnant Indian women in an earlier study (1). This is specifically important because B-12 levels progressively deteriorated through the trial in the placebo group. It was decided that it is unethical to allow deterioration in B-12 levels in the control group given the potential consequences of b-12 deficiency in infant and mother. The intervention dose of 250 mcg has been documented to cause biochemical improvement in B-12 status of pregnant women in Bangladesh (2). Hence for this trial specifically the 250 mcg is the intervention group which the 50 mcg is the quasi control group. This has been emphasised and clarified in the revised manuscript. (Page 4-5) The study seeks to test whether an intervention dose (250 mcg) will be able to improve the biochemical markers of B-12 deficiency through the duration of supplementation while the quasi control group only prevents further B-12 deterioration in the control group. Also, our

	study incorporates biochemical evaluation of B-12 status at baseline, mid term and post supplementation. Whether both groups improve, or one group improves (higher dose) will allow inference on efficacy of either or none or both. This biochemical change can then be correlated with the primary outcome variable i.e. neurodevelopment. This has been elaborated in the discussion section of the revised manuscript as suggested).
The direct and indirect biomarkers of B12 will be crucial in the interpretation of their results, particularly negative findings. The description of the laboratory procedures is weak. Please improve this section.	The biochemical analysis section has been described in greater detail as suggested in the revised version of the manuscript (Page.11)
They should mention the planned biomarkers for estimating iron status. At this stage, I believe that the study has already started enrolment, the authors should accordingly know more about how and where the samples are going to be analysed and provide more details about the preanalytical sample processing and storage. Please update the manuscript with this information.	The biomarkers for iron status are serum ferritin, serum iron and TIBC. This information has been added as suggested (Page.11). Pre-analytical sample processing and storage has been elaborated in the revised version of the manuscript (Page 9-10)
The table, "Table 2. Blood samples and measurements are incomplete and offer very little information about the biomarkers, biochemical methods, or sample. I am also unable to find a reference to this table in the text.	The information on biochemical is now detailed in the biochemical analysis section of the manuscript. Hence table 2 has been deleted.
Please be clear in the objectives, whether it is the DASII or BSID that will be the variable used to address the main objective.	DASII will be the primary toll for neurodevelopmental evaluation. This has been clarified in the revised version of the manuscript. (Page 11)
Several additional objectives are listed. Please refrain from using the term "effect" when referring to analyses that will be undertaken in an observational design.	Modified as Suggested (Page 5)
I suggest that you include a section that describes the chosen doses of B12 in relation to the recommended daily intakes.	The authors humbly submit that comparison with RDAs is likely to confuse the reader. RDA by definition (3) is the intake suggested for reference adults and not for those with pre-

	existing deficiency. In our case the vegetarian population is documented to be 70-90% (4) deficient and a progressive deterioration in deficiency has been shown in pregnancy. The dose used as quasi control has been shown to prevent this deterioration in deficiency in trial (1) and hence chosen. The study presumes on the basis of existing literature that the quasi control dose is only going to prevent deterioration in deficiency specifically in pregnancy. Comparisons with RDA would mean the quasi-control group to be 20-25 times the RDA and the intervention group to be 100 times the RDA. In effect this could be interpreted in a way that the study expects both the doses to be effective to a lesser or greater extent (High vs Higher dose comparison). This is not the primary hypothesis of the trial as now clarified in the revised version of the manuscript. (Page 4-5)
Please ensure that the information provided in this manuscript (especially inclusion criteria, exclusion criteria, objectives, and outcomes match the information that is given in the clinical trial registries.	The information provided has been re-verified with that provided in the registries and is found to be matching
Reconsider all the statements regarding strengths and limitations. This includes: “First large-scale, multi-centre randomised controlled, double-blind trial to determine efficacy of vitamin B-12 supplementation up to 6 months post-partum in improving neurodevelopmental outcomes in infancy.” There is a larger study with a similar objective going on in Nepal now the primary outcome is going to be measured at 9 months, why “up to 6 months post-partum” in this statement Minor issues Blood/plasma/serum “B12 levels” is very unscientific and imprecise. I would instead use the term “concentration” and “cobalamin.”	The strengths and limitations section have been revised in accordance with the observations of the reviewer (Page 3)

I would also use the term total homocysteine rather than “homocysteine.”	Modified as suggested
Metabolic screening, what is included in this assessment.	Metabolic Screening includes blood spot testing for the following- hypothyroidism, Galactaosemia..... The same has been detailed in the revised version of the manuscript (Page 10-11)
It is unclear to me whether or not the Bayley Scales of Infant development will be used or not. It is mentioned in the text and figure 1.	Only DASII scale will be used (The scale is adapted and validated from BSID). This has been clarified in the revised version of the manuscript) (Page 11)
Reconsider using reference 1 for gestational malnutrition	The reference has been replaced.
Reviewer: 2	
The study will only include vegetarian pregnant mothers. What is the rationale? What are the strengths/limitations of such a design? In my understanding, that the study is restricted to vegetarian pregnant women is such an important point of the study design, that it should be stated throughout the manuscript, including the title. What is the rationale for the vitamin B12 doses? Why compare to 50mcg and not a full placebo? What are strengths/limitations to such a design. Both points should be addressed more closely in the paper, starting from (title) the introduction, objectives, methods and discussion.	The paper has been modified to emphasise the points highlighted by the reviewer. Details are now included in the introduction on why vegetarian mothers were chosen and why the specific doses were chosen. (Page 4-5)

Introduction: The references used in the introductions seems outdated. There are more recent published reviews and studies on B12, perinatal health, children and cognition that would improve the introduction. Among these studies from New Delhi and the Kathmandu valley that should be very relevant for this study. Check also the literature on B12 and growth, both observational results from Nepal that are interesting for this study, and results from interventional trial in young children in Delhi.	Available literature on the subject has been thoroughly re-reviewed and some of the ongoing work has been added in the revised manuscript. (Page 4-5)
First sentence of third paragraph does not make completely sense, how can evidence be inconclusive when only one trial is referred to? Looks like there is some studies suggesting that there is a link, but very few (one?) trials have tried to confirm the effect of B12 on offspring neurodevelopment. In this trial however, the results do not provide evidence for a link between B12 and early child development. This could be due to dose? I would work some more on this paragraph.	The language of the paragraph has been modified to reflect the same in the revised version (Page 4)
Primary objective: include vegetarian.	Modified as suggested (Page 5)
Secondary objectives: please state which biochemical parameters.	Modified as suggested (Page 5)
Tertiary objectives; please reconsider the wording “effect” – will the assessment of causality be possible in these objectives, or will you look for associations?	Reworded as suggested (Page 5)
Methodology Again, it is important to include the point that you are including vegetarians only, under subheading “participants”: you state the number of births in Delhi and Kathmandu – how many of these	Clarified in in revised manuscript (Page 6)

mothers will be vegetarians? In my experience, not all.	
Procedure: check sentence line 20 page 12	Corrected in revised version (Page 13)
Neurodevelopment assessment: quality control procedures such as training, standardization and prevention of examiners drift should be detailed. How to make sure that testing in Nepal and India is done in a similar manner for the scores to be comparable? Will cultural adaptations be necessary when using the DASI in Nepal?	Details on Ensuring quality control are now added to the revised manuscript. To ensure that testing is done in similar manner in India and Nepal the Research staff from both India and Nepal will undergo joint training by a master trainer in India on DASII, followed by video-recording of assessments and determination of inter-observer variability. Cultural differences are not expected between India and Nepal although HOME environmental differences are expected. These will be assessed and documented.
The HOME: when, how and by whom? This information is lacking.	The information has been incorporated as suggested. (Page 11)
Statistics: Not clear to me how the subgroup analyses will be done? What are the subgroups. 2 subgroups, or 5 subgroups (bullet points), the sentence is not clear.	The analysis section has now been expanded to include these details.(Page 12-13)
Discussion: Some more on strength and limitations both of including vegetarians, and on not using a placebo control. Generalizability. What would be the implication for the findings? Implications of null findings?	We have reworked on the Discussion section and modified some sentences to incorporate these comments. (Page 15)
BSID III should be explained in a footnote. Please check the version, I think DASI is a version of the Bayley 2nd version, not the third. Also, I think you should also state DASI in the figure, not only BSID.	The error has been rectified. (Page 11)

References for the comments:

1. Duggan C, Srinivasan K, Thomas T, Samuel T, Rajendran R, Muthayya S, Finkelstein JL, Lukose A, Fawzi W, Allen LH, Bosch RJ. Vitamin B-12 supplementation during pregnancy and early lactation

increases maternal, breast milk, and infant measures of vitamin B-12 status. The Journal of nutrition. 2014 Mar 5;144(5):758-64.

2. Siddiqua TJ, Ahmad SM, Ahsan SB, Rashid M, Roy A, Rahman SM et al. Vitamin B12 supplementation during pregnancy and postpartum improves B12 status of both mothers and infants but vaccine response in mothers only: a randomised control in Bangladesh. Eur J Nutr. 2016; 55(1):281-293.

3. Allowances RD. Subcommittee on the Tenth Edition of the RDAs. Food and Nutrition Board, Commission on Life Sciences, National Research Council. Washington DC: National Academy Press. 1989.

4. Pathak P, Kapil U, Yajnik CS, Kapoor SK, Dwivedi SN, Singh R. Iron, folate, and vitamin B12 stores among pregnant women in a rural area of Haryana State, India. Food and nutrition bulletin. 2007 Dec;28(4):435-8.

VERSION 2 – REVIEW

REVIEWER	Tor Strand Sykehuset Innlandet Helseforetaket, Research
REVIEW RETURNED	30-Dec-2019

GENERAL COMMENTS	Thanks for allowing me to review this paper again. This is indeed a well-planned and necessary RCT, and I am looking forward to seeing the results from this effort. A well-written protocol paper that is aligned with other key documents (such as clinical trial registries) will substantially increase the impact of the observed results. I, therefore, suggest that the authors spend some time to sharpen the document and ensure that it is free from errors and inconsistencies. In my view, a protocol paper should be 1) particularly accurate about the background, 2) justify well the intervention, and 3) give a clear description of the objectives and methods. For various reasons, most of them mentioned below, and the submitted manuscript could be improved concerning all of these three aspects. In their updated manuscript, the authors have addressed many, but not all, of the issues raised from the first review. Also, some of their responses, such as the discussion of the implications of a null finding, are inaccurate. While other responses, such as the description of the studies included in the literature review, are incorrect (see below). Page 4: line 29: “Many observational studies detail the association between B-12 deficiency during pregnancy or early infancy with subsequent poorer infant neurodevelopment [14]” - Reference 14 does not cover this argument Page 5: line 3: Other RCTs that assessed a 250mcg dose of B-12 per day throughout pregnancy until 3 months’ postpartum in Bangladesh [22], and >2000mcg from 17 to 34 weeks’ gestation in India [23] did not evaluate neurodevelopment outcomes in infancy - Reference 23 is not a RCT. This Publication (Katre et al APJCN) was an observational study in 163 pregnant Indian women. Page 12, lines 47 onward:
--

- misspelling and abbreviations (CLIA, TIBC, CBC, RBC) that have not been explained earlier

Page 4:39 -44 Reconsider the following phrase. "An infant supplementation trial reported neurodevelopmental benefit of infant B-12 supplementation with folate [21]. Except for vitamin B12 acid alone on early child development. In another trial, maternal vitamin B-12 supplementation given"

Objectives:

Primary: Spell out the exact variables that will be measured. Not only "neurodevelopment" but mention the tool and domain that will be measured.

Secondary: Mention the biochemical variables that will be included in the "biochemical parameters"

Tertiary: I think that these can be spelled out even more clearly. The way they are phrased now, I am uncertain whether the parameters/variables will be used as predictors, mediators, or effect modifiers in the planned analyses.

Sample size calculations: A DQ score of 2 is the minimum difference that the authors do not want to overlook. This figure is meaningless without an expected SD. Please provide.

The authors provide the following response to my earlier question "The reviewer has correctly highlighted that biochemical evaluation was conducted in the Srinivasan trial. However, there was no improvement in maternal vitamin B-12 status from the first to the third trimester. In the supplemented arm increase was 3.0 (-54.8, 83.5) pmol/L, which was not significant. No significant group differences in maternal MMA, tHcy, or prevalence of anaemia (hemoglobin <11g/dL) were noted at the second or third trimester time points. "

- The numbers taken from Duggan's paper (Srinivasan trial) is correct but give a somewhat wrong impression of the results. The statement regarding group differences is incorrect. Please read the paper. In this RCT (Duggan et al. 2014), at baseline the median cobalamin was 160 pmol per liter, which increased to 216 in the 2 trimester and to 184 in the third trimester in the intervention group. There was also a significant group differences between the placebo and the 50µg cobalamin group: "Compared with the women who were administered placebo, vitamin B-12-supplemented women had significantly higher plasma vitamin B-12 concentrations in both the second [median vitamin B-12 concentration: 216 pmol/L (n = 119) vs. 112 pmol/L (n = 119), P < 0.001] and third [median: 184 pmol/L (n = 102) vs. 105 pmol/L (n = 102), P < 0.001] trimesters." Please also see figure 2 in Duggan's manuscript. Thus, the effect in cobalamin concentration was similar to the study in Bangladesh that used 250 µg. The response in the children at 6 weeks was excellent with regard to all measured biomarkers (cobalamin, MMA, and thcy)

Thus, the justification of using 250 ug is made from the response in the biochemical marker concentrations 3 days postpartum in the intervention group of the study in Bangladesh that used 250 µg. The two studies (Duggan and Siddiqua), which justifies this, represent very different populations, which is also reflected in the

	biomarker concentrations at baseline. Both studies showed a substantial effect of giving vitamin B12 on plasma concentration of cobalamin. However, this was not matched by a significant decrease in the tHcy concentrations in neither of the studies. In other words, the metabolic and biochemical response in the two RCT is not as different as the authors behind this protocol claim. In their revised manuscript, I suggest that they revise their description of the two studies and how null findings from this RCT will be interpreted. The dose discussion can also be improved by the findings from another recent B12 supplementation study in a similar (albeit not pregnant) population. (Yajnik 2019 PlosOne) The secondary and tertiary (or additional) objectives in the protocol paper are still not well aligned with those listed in clinicaltrials.gov (have not checked the Indian trial registry). I suggest that the authors straighten this up as these discrepancies will limit the publishability of the planned reports. Minor issues: Be consequent in the use of b-12 and B-12 (capitalization) Lab; page 10: line 59: Sample processing: only providing RPM without rotor radius or diameter of the centrifuge does not make sense. Page 4, line 28, "Sufficient B-12 intake during pregnancy is vital for foetal brain development.": -Needs reference
--	--

REVIEWER	Ingrid Kvestad Regional Centre for Child Mental Health and Welfare, NORCE Norwegian Research Centre, Bergen, Norway
REVIEW RETURNED	11-Dec-2019

GENERAL COMMENTS	Thank you for this opportunity to review the paper for the second time. I find the paper to be improved, still I have some minor suggestions: The paper would benefit from a critical review of the English language. Study strengths Page 3 line 23, Include a wording indicating that the study is on maternal supplementation during pregnancy/after birth. Introduction page 5 lines 24 to 27: Please provide some reference to the claims in this sentence. On the DASII Page 12 lines 15-22: Please re-check the BSID version that the DASII is adapted from. I am not able to access reference 30. The test that the authors describe in these lines, is not the 3rd version, but the Bayley scales of Infant and Toddlers development 2nd edition (BSID-II). The second version has two scales (as described in this paper), the third has three scales (cognitive, language and motor). The third version is currently used for research in Nepal and the feasibility was evaluated in publication
--

	by Ranjitkar and colleagues published in Frontiers of Psychology earlier this year. On Biochemical analyses page 12 lines 47 and onwards There are inconsistencies in how the biomarkers are described from the mothers to the infants (i.e. B-12 – Vitamin b-12 – Vitamin B-12 and total homocysteine – Homocysteine and 25 (OH)D – Vitamin D – holo-transcobalamin and Holotranscobalamin). These should be consistent in this paragraph and please also check the introduction and the paper as whole. Discussion page 16 line 38: Capital letter in new sentence Discussion page 16 lines 45 and onwards: I suggest that the authors elaborate more on how a negative results or null findings will "add to existing knowledge and lay the groundwork for future work on the subject especially around utility, dose, duration and time of initiation of Vitamin B12 supplementation in pregnant women/infants." As pointed out by the reviewers in the first round, the use of a quasi-control group receiving 50 mcg daily will represent a challenge in interpreting the results in case of a null-finding. In my view, the use of quasi-control group, although perhaps required ethically since the study target vegetarian women at increased risk of more severe vitamin B12 deficiency, is a limitations to the study and negative and null findings will be very challenging to interpret, i.e. what is the benefit of 50 mcg compared to no B12? I think this needs to be addressed in the discussion.
--	--

VERSION 2 – AUTHOR RESPONSE

Reviewers Comments	Author Response
Reviewer: 1 Reviewer Name: Tor Strand Institution and Country: Innlandet Hospital Trust, Norway	
Page 4: line 29: “Many observational studies detail the association between B-12 deficiency during pregnancy or early infancy with subsequent poorer infant neurodevelopment [14]” - Reference 14 does not cover this argument	The referencing has been rechecked and rectified.
Page 5: line 3: Other RCTs that assessed a 250mcg dose of B-12 per day throughout pregnancy until 3 months’ postpartum in Bangladesh [22], and >2000mcg from 17 to 34 weeks’ gestation in India [23] did not evaluate neurodevelopment outcomes in infancy - Reference 23 is not a RCT. This Publication (Katre et al APJCN) was an	The error has been rectified.

observational study in 163 pregnant Indian women.	
Page 12, lines 47 onward: - misspelling and abbreviations (CLIA, TIBC, CBC, RBC) that have not been explained earlier	Page 12, lines 47 has now been modified as suggested by the reviewer.
Page 4:39 -44 Reconsider the following phrase. “An infant supplementation trial reported neurodevelopmental benefit of infant B-12 supplementation with folate [21]. Except for vitamin B12 acid alone on early child development. In another trial, maternal vitamin B-12 supplementation given”	Clarified as suggested (Page 4)
Objectives: Primary: Spell out the exact variables that will be measured. Not only “neurodevelopment” but mention the tool and domain that will be measured.	Clarified as suggested (Page 5)
Secondary: Mention the biochemical variables that will be included in the “biochemical parameters”	Clarified as suggested (Page 5)
Tertiary: I think that these can be spelled out even more clearly. The way they are phrased now, I am uncertain whether the parameters/variables will be used as predictors, mediators, or effect modifies in the planned analyses.	Clarified as suggested (Page 5-6)
Sample size calculations: A DQ score of 2 is the minimum difference that the authors do not want to overlook. This figure is meaningless without an expected SD. Please provide.	The information has been incorporated as suggested. (Page 13)

The authors provide the following response to my earlier question “The reviewer has correctly highlighted that biochemical evaluation was conducted in the Srinivasan trial. However, there was no improvement in maternal vitamin B-12 status from the first to the third trimester. In the supplemented arm increase was 3.0 (-54.8, 83.5) pmol/L, which was not significant. No significant group differences in maternal MMA, tHcy, or prevalence of anaemia (hemoglobin <11g/dL) were noted at the second or third trimester time points. “

- The numbers taken from Duggan’s paper (Srinivasan trial) is correct but give a somewhat wrong impression of the results. The statement regarding group differences is incorrect. Please read the paper. In this RCT (Duggan et al. 2014), at baseline the median cobalamin was 160 pmol per liter, which increased to 216 in the 2 trimester and to 184 in the third trimester in the intervention group. There was also a significant group differences between the placebo and the 50µg cobalamin group: “Compared with the women who were administered placebo, vitamin B-12-supplemented women had significantly higher plasma vitamin B-12 concentrations in both the second [median vitamin B-12 concentration: 216 pmol/L (n = 119) vs. 112 pmol/L (n = 119), P < 0.001] and third [median: 184 pmol/L (n = 102) vs. 105 pmol/L (n = 102), P < 0.001] trimesters.” Please also see figure 2 in Duggan’s manuscript. Thus, the effect in cobalamin concentration was similar to the study in Bangladesh that used 250 µg. The response in the children at 6 weeks was excellent with regard to all measured biomarkers (cobalamin, MMA, and thcy)

Thus, the justification of using 250 ug is made from the response in the biochemical marker concentrations 3 days postpartum in the intervention group of the study in Bangladesh that used 250 µg. The two studies (Duggan and Siddiqua), which justifies this, represent very different populations, which is also reflected in the biomarker concentrations at baseline. Both studies showed a substantial effect of giving vitamin B12 on plasma concentration of cobalamin. However, this was not matched by a significant decrease in

While we agree with the reviewer we would like to humbly submit that the protocol is based on the possibility that the 250 mcg B-12 supplementation might be better than 50 mcg. In our reading of the results published by Duggan et al there is a possibility that the dose of 50 mcg was inadequate and/or given for a shorter duration. This interpretation is consistent with the views expressed by another reviewer. It is definitely not the only interpretation and the views of the reviewer could definitely be correct. The best way to answer the question would be a head to head comparison of the two doses and hence this trial is proposed. To further detail the reason for this possibility we quote the results from the Duggan et al paper.

“Compared with the women who were administered placebo, vitamin B-12-supplemented women had significantly higher plasma vitamin B-12 concentrations in both the second [median vitamin B-12 concentration: 216 pmol/L (n = 119) vs. 112 pmol/L (n = 119), P < 0.001] and third [median: 184 pmol/L (n = 102) vs. 105 pmol/L (n = 102), P < 0.001] trimesters. The median

change in maternal vitamin B-12 status from the first to third trimester in the supplemented arm was 3.0 (254.8, 83.5) pmol/L, and this was significantly different (P < 0.001) from the decline in the placebo arm, 237.6 (266.5, 210.9) pmol/L. The median change in maternal vitamin B-12 status from the first to second trimester in the supplemented arm was 36.5 (22.6, 50.2) pmol/L, and this was significantly different (P < 0.001) from the decline

in the placebo arm, 246.3 (266.7, 225.9) pmol/L. No significant group differences in maternal MMA, tHcy, or prevalence of anaemia (haemoglobin < 11 g/dL) were noted at the second or third trimester time points.”

We largely agree that there is evidence of biochemical improvement in the mother and infant. However, if we look at the change in biochemical parameters during supplementation (generally considered better than direct time point group comparisons) it is evident there is significant and consistent decline in B-12 concentrations in the placebo group. This could be explained by higher requirements during pregnancy. In the supplemented group the mean change between first to third trimester 3.0 (254.8, 83.5) pmol/L which is statistically not significant.

the tHcy concentrations in neither of the studies. In other words, the metabolic and biochemical response in the two RCT is not as different as the authors behind this protocol claim. In their revised manuscript, I suggest that they revise their description of the two studies and how null findings from this RCT will be interpreted.

Also, no significant group differences in maternal MMA, tHcy, or prevalence of anemia (haemoglobin < 11 g/dL) were noted at the second or third trimester time points. Given the peculiarity/inconsistency one possible interpretation is that the 50-mcg dose was causing a marginal rise and mainly mitigating the fall that was happening in the placebo group. To our understanding in a population which is severely deficient (prevalence of deficiency is up to 74 %), is vegetarian and pregnant just mitigating the pregnancy related fall or marginal improvement may not be enough for desirable outcomes and hence proposed the study currently in progress. This interpretation is inspired by a review by Siddiqua et al where he says "A recent randomized study among Indian women reported that supplementation with B12 (50 µg/day) throughout pregnancy up to 6 week postpartum increased the concentration in maternal and infant plasma and breast milk B12 content [85]. However, both the dose and the duration of supplementation are important to sustain optimum B12 status in circulation in mother-infant pairs. The observations by Duggan et al should be replicated by well-designed RCTs with optimum dose and extended beyond their 6 wk postpartum observation period." He subsequently goes on to the use of 250 mcg in their pilot study.

The language of the modified manuscript has been revised to reflect the possibility discussed above while refraining from making any claims on interpretation/conclusions from Duggan et al or Siddiqua et al.

If there are null findings (neurodevelopment comparable between two groups)) from the current study, they will be interpreted as follows

1. Findings: Improvement in B-12 parameters from first trimester to third trimester is statistically significant but the change is statistically comparable between two groups for mother and baby. Substantial proportion of mother/infants in both groups improve from B-12 deficient to replete state.

Interpretation: 250 mcg dose is as effective as 50 mcg in improving biochemical parameters and both doses result in comparable neurodevelopment.

	250 mcg dosing offers no additional advantage over 50 mcg. 2. Findings: Improvement in B-12 parameters from first trimester to third trimester is significant in either 250 mcg alone or both groups but 250 mcg causes higher rise (statistically significant) in B-12 biochemical parameters than 50 mcg for mother and baby. Interpretation: 250 mcg is more efficacious than 50 mcg in improving biochemical B-12 status. Since neurodevelopment is equivalent between two groups a higher dose of supplementation does not offer any neurodevelopmental advantage despite better biochemical B-12 status. 3. Findings: A lack of biochemical improvement in both groups although unexpected would be interpreted to mean poor efficacy of 50 mcg or 250 mcg oral B-12 in improving B-12 status of mother infant pairs. Both groups continue to have high proportion of mother deficient in B-12 before and after supplementation. Interpretation: Factors affecting B-12 status in the current trial (including compliance and socio-demographics) would have to carefully analysed. Further work would be required to study the determinants of oral B-12 efficacy in detail.
The dose discussion can also be improved by the findings from another recent B12 supplementation study in a similar (albeit not pregnant) population. (Yajnik 2019 PlosOne)	The use of physiological dose by Yajnik et al (3) in adolescent girls has now been taken up in the Discussion section of the revised version of the manuscript. (Page 16-17)
The secondary and tertiary (or additional) objectives in the protocol paper are still not well aligned with those listed in clinicaltrials.gov (have not checked the Indian trial registry). I suggest that the authors	The outcomes provided (secondary and additional) have been modified in the registries and is verified to be matching. (Page 5-6)

straighten this up as these discrepancies will limit the publishability of the planned reports.	
Minor Issues: Be consequent in the use of b-12 and B-12 (capitalization) Lab; page 10: line 59: Sample processing: only providing RPM without rotor radius or diameter of the centrifuge does not make sense. Page 4, line 28, “Sufficient B-12 intake during pregnancy is vital for foetal brain development.”: -Needs reference	The error has been rectified. (Page 12) Rotor radius: 15 cm. The Same has been incorporated in the revised version of the manuscript (Page 10) Reference has been added as suggested. (Page 4)
Reviewer: 2 Reviewer Name: Ingrid Kvestad Institution and Country: Regional Centre for Child Mental Health and Welfare, NORCE Norwegian Research Centre, Bergen, Norway	
The paper would benefit from a critical review of the English language.	The language of the paper has been critically reviewed in the revised version
Study strengths Page 3 line 23, Include a wording indicating that the study is on maternal supplementation during pregnancy/after birth.	The strengths section has now been revised in accordance with the observations of the reviewer. (Page 3)
Introduction page 5 lines 24 to 27: Please provide some reference to the claims in this sentence.	References has been added as suggested. (Page 5)
On the DASII Page 12 lines 15-22: Please re-check the BSID version that the DASII is adapted from. I am not able to access reference 30. The test that the authors describe in these lines, is not the 3rd version, but the Bayley scales of Infant and Toddlers development 2nd edition (BSID-II). The second version has	The error has been rectified. (Page 11). The pdf of reference 30 has been uploaded as an additional file

two scales (as described in this paper), the third has three scales (cognitive, language and motor). The third version is currently used for research in Nepal and the feasibility was evaluated in publication by Ranjitkar and colleagues published in Frontiers of Psychology earlier this year.	
On Biochemical analyses page 12 lines 47 and onwards There are inconsistencies in how the biomarkers are described from the mothers to the infants (i.e. B-12 – Vitamin b-12 – Vitamin B-12 and total homocysteine – Homocysteine and 25 (OH)D – Vitamin D – holotranscobalamin and Holotranscobalamin). These should be consistent in this paragraph and please also check the introduction and the paper as whole.	Page 12, lines 47 onwards has now been modified as suggested by the reviewer. (Page 12)
Discussion page 16 line 38: Capital letter in new sentence	The sentence has been removed in the revised version to accommodate the comments of Reviewer 1.
Discussion page 16 lines 45 and onwards: I suggest that the authors elaborate more on how a negative results or null findings will "add to existing knowledge and lay the groundwork for future work on the subject especially around utility, dose, duration and time of initiation of Vitamin B12 supplementation in pregnant women/infants." As pointed out by the reviewers in the first round, the use of a quasi-control group receiving 50 mcg daily will represent a challenge in interpreting the results in case of a null-finding. In my view, the use of quasi-control group, although perhaps required ethically since the study target vegetarian women at increased risk of more severe vitamin B12 deficiency, is a limitations to the study and negative and null findings will be very challenging to interpret, i.e. what is the benefit of 50 mcg compared to no B12? I think this needs to be addressed in the discussion.	The interpretation and the limitations arising out of the same have been incorporated in the revised version of the manuscript If there are null findings (neurodevelopment comparable between two groups)) from the current study, they will be interpreted as follows  4. Findings: Improvement in B-12 parameters from first trimester to third trimester is statistically significant but the change is statistically comparable between two groups for mother and baby. Substantial proportion of mother/infants in both groups improve from B-12 deficient to replete state. Interpretation: 250 mcg dose is as effective as 50 mcg in improving biochemical parameters and both doses result in comparable neurodevelopment. 250 mcg dosing offers no additional advantage over 50 mcg. 5. Findings: Improvement in B-12 parameters from first trimester to third trimester is significant in either 250 mcg alone or both groups but 250 mcg causes higher rise (statistically significant) in B-12 biochemical

	parameters than 50 mcg for mother and baby. Interpretation: 250 mcg is more efficacious than 50 mcg in improving biochemical B-12 status. Since neurodevelopment is equivalent between two groups a higher dose of supplementation does not offer any neurodevelopmental advantage despite better biochemical B-12 status. 6. Findings: A lack of biochemical improvement in both groups although unexpected would be interpreted to mean poor efficacy of 50 mcg or 250 mcg oral B-12 in improving B-12 status of mother infant pairs. Both groups continue to have high proportion of mother deficient in B-12 before and after supplementation. Interpretation: Factors affecting B-12 status in the current trial (including compliance and socio-demographics) would have to carefully analysed. Further work would be required to study the determinants of oral B-12 efficacy in detail.
--	--

References:

1. Duggan C, Srinivasan K, Thomas T, Samuel T, Rajendran R, Muthayya S, Finkelstein JL, Lukose A, Fawzi W, Allen LH, Bosch RJ. Vitamin B-12 supplementation during pregnancy and early lactation increases maternal, breast milk, and infant measures of vitamin B-12 status. *The Journal of nutrition*. 2014 May 1;144(5):758-64.
2. Siddiqua JT. Vitamin B12 deficiency in pregnancy and lactation: is there a need for pre and post-natal supplementation? *J Nutr Disord Ther*. 2014;4(2):1-8.
3. Yajnik CS, Behere RV, Bhat DS, Memane N, Raut D, Ladkat R, Yajnik PC, Kumaran K, Fall CH. A physiological dose of oral vitamin B-12 improves hematological, biochemical-metabolic indices and peripheral nerve function in B-12 deficient Indian adolescent women. *PLoS one*. 2019;14(10).

VERSION 3 – REVIEW

REVIEWER	Tor Strand University of Bergen
REVIEW RETURNED	08-Mar-2020
GENERAL COMMENTS	The authors argues well for their choice of approach and the introduction reads much better now.

	The authors has included a section on how to interpret negative results which is good. However, I suggest that they improve the language in this section. Line 52, page 4. Please make clear that reference 24 is an observational study. There are also some errors in the references, please check
--	---

REVIEWER	Ingrid Kvestad Regional Centre for child and youth mental health and welfare, NORCE Norwegian Research Centre, Bergen, Norway
REVIEW RETURNED	08-Mar-2020

GENERAL COMMENTS	Thank you again for letting me review this protocol paper. The paper has been considerably improved, the literature and references in the introduction seems more up to date and my other comments have mostly been taken into consideration. Minor comments: I recommend a critical review and proof read of the paper. The flow of the language is still not adequate. Also there are small inconsistencies throughout the paper, for instance in the introduction: page 6 line 22 - reference number 22 is not right in this context and should be switched (perhaps with reference 23 which is a study in pregnant women?). I saw this by chance, the other references should also be checked. The use of capital letters and not is inconsistent. Please also have a close look at the primary objective. I recommend to remove the parenthesis on the quotient of the DASII. The primary objective should be very clear, please remember that future reviewers may look into this paper to see if the analyses have been carried out according to the plan. Is it one quotient or two (add an s?) - I think the latter is correct since you later on in the paper state that the DASII has a motor scale and a mental scale. I wish you good luck!
---

VERSION 2 – AUTHOR RESPONSE

Reviewers Comments	Author Response
Reviewer: 1 Reviewer Name: Tor Strand Institution and Country: Innlandet Hospital Trust, Norway	
The authors argues well for their choice of approach and the introduction reads much better now. The authors has included a section on how to interpret negative results which is good. However, I suggest that they improve the language in this section.	The language for the concerned section has been reviewed and changed in the revised Version. The referencing has been rechecked and rectified. (Page 4)

Line 52, page 4. Please make clear that reference 24 is an observational study. There are also some errors in the references, please check	
Reviewer: 2 Reviewer Name: Ingrid Kvestad Institution and Country: Regional Centre for Child Mental Health and Welfare, NORCE Norwegian Research Centre, Bergen, Norway	
Thank you again for letting me review this protocol paper. The paper has been considerably improved, the literature and references in the introduction seems more up to date and my other comments have mostly been taken into consideration. Minor comments: I recommend a critical review and proof read of the paper. The flow of the language is still not adequate. Also there are small inconsistencies throughout the paper, for instance in the introduction: page 6 line 22 - reference number 22 is not right in this context and should be switched (perhaps with reference 23 which is a study in pregnant women?). I saw this by chance, the other references should also be checked. The use of capital letters and not is inconsistent.	The language of the paper has been critically reviewed in the revised Version. The referencing has been rechecked and rectified. (Page 6)
Please also have a close look at the primary objective. I recommend to remove the parenthesis on the quotient of the DASII. The primary objective should be very clear, please remember that future reviewers may look into this paper to see if the analyses have been carried out according to the plan. Is it one quotient or two (add an s?) - I think the latter is correct since you later on in the paper state that the DASII has a motor scale and a mental scale. I wish you good luck!	The primary objective has now been modified in the manuscript.